# The Effect of Humidity on the Atomization Process and Structure of Nanopowder Designed for Extinguishment

**DOI:** 10.3390/ma14123329

**Published:** 2021-06-16

**Authors:** Mateusz Biel, Piotr Izak, Krystian Skubacz, Agata Stempkowska, Joanna Mastalska-Popławska

**Affiliations:** 1Faculty of Materials Science and Ceramics, AGH University of Science and Technology, Mickiewicza 30 Av., 30-094 Krakow, Poland; bielm@agh.edu.pl (M.B.); izak@agh.edu.pl (P.I.); jmast@agh.edu.pl (J.M.-P.); 2Silesian Centre for Environmental Radioactivity (BCR), Central Mining Institute, Plac Gwarków 1, 40-166 Katowice, Poland; kskubacz@gig.eu; 3Faculty of Civil Engineering and Resource Management, AGH University of Science and Technology, Mickiewicza 30 Av., 30-094 Krakow, Poland

**Keywords:** superfine nanopowder structure, extinguishing efficiency, active burning radicals, humidity

## Abstract

Increasingly, firefighting aerosols are being used to extinguish fires. It is assumed that the extinguishing mechanism involves breaking the chain of physicochemical reactions occurring during combustion by binding free radicals at ignition. The radicals are most likely formed from the transformation of water molecules, with the active surfaces of aerosol micro- or even nanoparticles. The aerosol extinguishing method is very effective even though it does not reduce oxygen levels in the air. In contrast to typical extinguishing powders, the aerosol leaves a trace amount of pollutants and, above all, does not adversely affect the environment by depleting the ozone layer and increasing greenhouse effects. Depending on how the firefighting generators are released, the aerosol can act locally or volumetrically, but depending on environmental conditions, its effectiveness can be variable. The article presents the influence of environmental humidity on the atomization of aerosol nanosize, which confirms the radical combustion mechanism. This paper presents the effect of environmental humidity on the atomization of aerosol superfine (nano) particles. The main focus was on the grain distribution and its effect on the surface activity of the FP-40C type firefighting aerosol. Changes in the characteristic parameters of the particle size distribution of RRSB (Rosin-Rammler-Sperling-Bennet) are presented.

## 1. Introduction

The mechanism of action of extinguishing spray is not yet well understood. It is assumed to be associated with the inhibition of the combustion reaction. In further explaining this mechanism, it is important to know the form of fuel, oxygen and the environment in which, during a fire, the chain reactions can occur. For simplicity, we can assume that coal and hydrogen are considered as fuels in the environment. Fuel can exist in different states of matter, however, and in the case of a flame combustion reaction, only the gaseous state carrying the mentioned hydrogen and carbon atoms, is present. A similar combustion mechanism may occur in the case of liquid fuel combustion, although in this case the liquid vaporizes first. In other words, it should be assumed that flame combustion occurs only in a gaseous environment. A schematic of the flame combustion of solid and liquid fuels is shown in Figure 1.

Heat is required to initiate radicals and to sustain pyrolysis or fuel evaporation. For feedback to occur as in Figure 1, it is necessary for chain reactions to take place during combustion. It turns out that the chain reactions of hydrogen and carbon oxidation occur with water. This is because water can produce active oxidizing radicals OH*, H*, O*, H_3_O*, H_2_O_2_*, etc., which support combustion reactions. Through radical recombination, the eventual fire is sustained until the potential for radical formation is exhausted [1,2,3].

An important role in this mechanism is performed by the presence of nitrogen, especially at temperatures >500 °C, both in the air (Zeldovich mechanism) and in the fuel (oxidation and reduction mechanisms) or as so-called fuel radicals according to the scheme in Figure 2. The contribution of nitrogen involves the activation of the oxygen radical O* [4,5,6].

On the other hand, for solid and liquid fuels, in the presence of oxygen and hydrogen, some intermediate gaseous compounds can be formed which undergo further oxidation and reduction reactions. For example, for coal, the total reaction is:2C + 3H_2_ +4O_2_ + initiator = 2CO_2_ + 4OH* + 2H*(1)

The text in bold indicates active water elements (radicals) that do not exist under natural conditions, or exist for very short periods of time in the order of picoseconds [7,8,9]. A removal or permanent binding of only one type of the radicals should slow down or break the chain reaction of combustion. If this is the case then fuel nomenclature should be redefined. The presence of combustion reaction radicals is associated with the unusual properties of water, which is the only one in nature that exists in three states of aggregation and participates in all natural phenomena. Nevertheless, to this day its structure is unknown [10,11,12,13,14,15,16]. Tokushima et al. have shown that water forms networks in which water molecules are bound by hydrogen bonds (dimers) to only two neighbors, rather than four neighbors as previously thought, as in ice [17]. This implies that the chains and/or rings formed by various forms of donorization are the most likely structures in the liquid water at temperatures between 7 °C and 100 °C, and that the basic “unit” is the hydronium ion H_3_O^+^ (oxonium), which at higher temperatures >100 °C “breaks down” relatively easily into H_2_O and H*. There is a lack of detailed data on the mechanism of this decay, although various suggestions can be found about the leading role of the 1b1’ orbital in electron supply at temperatures above 100 °C [3,18]. Thus, the total Equation (2) in the nitrogen environment is:C + H_3_O* + OH* + 2NO* = CO_2_ + 2H_2_O + N_2_(2)

Thermal dissociation (thermolysis) of “pure” water occurs at temperatures >2000 °C [19,20]; however, in catalytic and ionized environments, i.e., in the presence of salts or aerosols, this reaction can occur at much lower temperatures [21,22]. In practice, this is done by binding oxygen with a reducing agent. Alkali metal salts and elements from the 17th group of the periodic table work well in this role as reducers of the reaction during flame combustion. Unfortunately, halogenated volatile organic compounds (VOC) deplete the ozone layer and contribute to the global warming effect and can no longer be used to extinguish fires [23].

Extinguishing aerosols and powders work differently. They have a very active surface to support oxidation and reduction reactions. It is suggested that the radicals formed during reduction, e.g., OH*, can be recombined on the surfaces of micro or nanoparticles. Currently manufactured extinguishing aerosols based on potassium carbonates and nitrates have grain sizes in the range of 200 nm, which at aerosol release temperatures (>1000 °C) can be active centers immediately reacting with radicals that sustain the flame combustion reaction [24,25,26].

In recent years, there has been a clear increase in fires (13%). Among other things, climate change makes fires bigger, more intense and last longer than before. On the other hand, it is estimated (FAO) that humans are responsible for around 75% of all wildfires, and much of the increase in fire incidents during 2020 can be directly linked to human actions, both in terms of new technologies, materials and interventions. A science-based approach is needed to forecast risk and to prioritize interventions, which are both critical elements in preventing fires before they need to be suppressed.

Extinguishing powders have been used in firefighting for many years. The term is defined as all finely ground (non-flammable) compounds of inorganic salts surrounded by a hydrophobic membrane, designed to isolate burning bodies from the access of oxygen. Aerosol-extinguishing systems are not yet popular but have obvious advantages over other extinguishing methods due to their low toxicity, environmental protection, economic aspect and high efficiency [27]. With the large-scale use of firefighting powders and aerosols, the requirements for performance improvement are becoming higher and higher and many studies have been conducted on their quality and properties. Among other things, it was found that commercial dry powder, further processed to ultra-fine powder (particle size ≤20 μm), significantly affects the application properties [28,29]. An increased powder-specific surface area improves the interaction of the extinguishing agent with the flame [30] as does channel system porosity [31]. Unfortunately, with such developed surfaces, there are problems with the agglomeration of powder microparticles, especially aerosols. The chemical composition of powders is not without significance [31,32]. Worldwide research in this aspect generally deals with the effect of alkali metal cations on quenching efficiency [33,34]. The flame suppression effectiveness of powders may be a function of the developed surface area of the solid and the type of chemical reaction with their decomposition elements. For “nanosized” particles with a significantly increased surface area, higher fire suppression efficiencies are observed than for particles of a relatively larger size. However, in firefighting practice, there is a limit to the amount of powder particles that can be used in firefighting equipment. Associated with the reduction in particle size is particle agglomeration and, therefore, there is difficult contact between the particle surface and the flame radicals So far, the grain size distribution of superfine powder in relation to air humidity has not been studied. The primary objective of the research presented in this article was to see how the humidity of the fire environment affects the atomization of the extinguishing powder. Obtaining the smallest possible particles in the highest possible concentrations during the release of the extinguishing substance is a key element in successful firefighting.

## 2. Materials and Methods

### 2.1. Mechanism of Aerosol Action

However, the effectiveness of locally acting extinguishing aerosols depends on many spatial factors that are usually considered in the design of the extinguishing system FES (fixed extinguishing system) [35,36,37]. However, environmental factors such as humidity have not been analyzed so far. Previous research shows [1] that the flame is extinguished in the first seconds of aerosol release from aerosol generators. The aerosol probably reacts with combustion products and moisture to form a microaerosol in the form of a visible mist. After about 5 s, the flame in the closed test chamber is extinguished and after 15 s the mist is no longer visible due to evaporation (Figure 3). This fact suggests that the aerosol superfine powders can react with moisture and radicals of the combustion reaction. Hence, an extended knowledge of the flame combustion mechanism is presented, which as yet is not widely known, and may contribute to finding more effective extinguishing agents, e.g., extinguishing lithium batteries and the chemical assembly of oxidizing free radicals into molecules that can occur with the active surfaces of powder grains is well known. This effect, called heterophase inhibition, also occurs to varying degrees with typical powders designed for flame suppression. Although, in that case, other mechanisms are mostly used (heat absorption, oxygen limitation, etc.) [38].

In other words, it is suggested that the free radicals necessary for flaming combustion are captured and bound for a period of time by active sites on the outer surface of the powder. With radicals trapped in this way, other free radicals can react, forming larger molecules but with less activity. Hence, from a chemical inhibition point of view, typical powder extinguishers are less effective. But the areas of lower activity on the surface of the powder grains can also play a role in limiting combustion processes.

Other reactions may also take place in the flame. For example, a pyrotechnic or other charge of special composition, with rapid combustion and, above all, the absence of toxic components, allows effective atomization of alkaline substances, such as potassium bicarbonate. Suitably cooled and filtered aerosol, which is formed during the combustion of the pyrotechnic charge, lifts the salt nanoparticles and simultaneously causes their rapid decomposition to potassium oxide. The aerosol reaching the flame can therefore react very quickly with free radicals (homophase inhibition) due to its very large specific surface area. This is essentially an interruption of the chain reaction and a limitation of radical recombination due to the strong reducing properties of alkali ions [39]. Another aspect of extinguishing aerosol performance concerns the effect of humidity in the flame combustion environment on the atomization of the extinguishing aerosol. For this purpose, measurements have been carried out in a special chamber allowing to measure the concentration and size distribution of superfine particles in areosol FirePro^®^ FPC40 (Nuuxe) (proprietary composition) depending on the different humidity of the environment and thus to determine their surface activity of atomization. The fire extinguishing aerosol FirePro^®^ is also used in protecting valuable museum objects and is protected by patent application PL 422,204 A1 “Ref. [40] Application of fire extinguishing aerosol for protection of museum pieces and objects of historical interests against ageing and fire action”.

Nuuxe’s FirePro^®^ products contain the latest generation of our FPC aerosol material, which is mainly composed of potassium salts without any pyrotechnics. When activated, the FPC transforms into a rapidly expanding, highly effective extinguishing aerosol. As a result of the electric impulse, the extinguishing agent is activated and spontaneously discharged. The generated aerosol spreads through space as a result of its own momentum [41].

### 2.2. Advanced Measuring Methodology

Aerosol size distributions were measured in an isolated chamber with a volume of 17 m^3^ [39] using two particle spectrometers (Figure 4). Both spectrometers are products of TSI Instruments (Shoreview, MN, USA). The first one, the Scanning Mobility Particle Sizer (SMPS model 3936), is designed to measure particle concentration in the range of about 10 to 1000 nm. The measurement range depends on the flow rate.

The relative humidity in the chamber was changed in the range of 20–90%. During the tests, the aerosol flow rate was 0.3 dm^3^/min, which allowed recording particles in the range of 10–700 nm divided into 107 measurement channels. The SMPS spectrometer consists of three main units: an electrostatic classifier (EC), a differential mobility analyzer (DMA), and a condensation particle counter (CPC). At the aerosol inlet of the first unit there is a single-stage impactor (filter) which removes particles outside the measurement range. Inside the EC panel there is a column with a beta radiation source (Kr-85: β^+^), which ionizes the flowing aerosol and the particles to quickly reach a state of charge equilibrium. Inside the DMA column, aerosol-containing air combines with a ten times stronger stream of aerosol-cleaned air to maintain laminar flow along the column axis.

At the center of the column there is a negatively charged electrode that produces an electrostatic field (Figure 5). As a result, the trajectory of the positively charged particles is curved toward the column axis. If the trajectory curve is appropriate, the particles can reach the hole located underneath the column. During each measurement cycle, the electrode voltage is gradually changed so that particles of different sizes reach the hole and are directed to the CPC counter, where they are detected with the help of a laser. Before reaching the laser system, particles travel along the high humidity area to grow to a size where they can be detected. The spectrometer software allows assessment of aerosol losses inside the conducting tubes, which are dependent on temperature, air pressure, and capillary length. However, such losses are significant for particles, below 10 nm due to their higher diffusion coefficient. For larger particles, diffusion is not significant.

The second Aerodynamic Particle Sizer (APS model 3321) spectrometer is designed to capture larger particles. Its measuring range, divided into 51 channels, is from 542 to 19,810 nm. Air enters the spectrometer at a rate of 5 dm^3^/min. The air stream is divided into two parts. The stronger stream of about 4 dm^3^/min is cleaned by filters and, close to the two-laser system, combines with the air containing aerosols. As a result, the aerosols in the spectrometer accelerate and the transit time between the two lasers depends on their inertia. 

Each single measurement cycle, a scan of the total available number of grains for the APS and SMPS spectrometer, took 120 s. After a pause of 60 s, the measurement cycle was repeated until approximately 14 min had elapsed. The aerosol distributions obtained were described by commonly used parameters such as Count Media Diameter (CMD), Count Average Diameter (CAD), and Activity Average Diameter (AAD). These can be used to compare particle distributions after different times since aerosol atomization. Median is the mean grain diameter (CMD) dividing the distribution into two equal parts. The mode determines the most frequent fraction of nano-grains (MOD). The mean value of the nano-grains (CAD) refers to the whole population, which is 36.8% when the Rosin-Rammler-Sperling-Bennett (RRSB) frequency function is satisfied.

## 3. Results

The size distribution of the nanoparticles after atomization in different moisture environments and also at different time intervals is shown in Figure 6, Figure 7, Figure 8, Figure 9 and Figure 10 with the population values as the average, median, and mode (dominant). First of all, the shape of the population distribution curves after atomization changes with time. The study shows that a high concentration of aerosol particles occurs at the beginning of atomization, i.e., during the first 120 s of measurement, which decreases rapidly with time. This is illustrated in Figure 6, Figure 7, Figure 8, Figure 9 and Figure 10, showing the nanoparticle counts of the most common fraction (MOD).

However, at a relative humidity of 90% and both 30 and 40%, a pronounced flattening of the most common fractions can also be observed. Overall, after 10 min, the modal fraction count stabilizes at 25,000 grains. However, at a relative humidity of 30 to 90%, an increase in grain size of the most common fraction (MOD) by about 50 nm can be observed within about 10 min, with different levels of increase for dry (<50%) and wet (>50%) environments (Figure 11). It is likely that at high humidities, the nanoparticles in the measuring chamber clump together volumetrically, whereas at low humidities, they aggregate.

It is likely that at a high humidity, volumetric agglomeration (coagulation) of the nano-grains in the measuring chamber has already started occurring. The higher the humidity of the environment, the greater the number of smaller nanoparticles (this dependence is particularly true up to a humidity of 70%; at 90% the number drops sharply). The average diameter of the population nanoparticles measured at the same times after atomization decreases by about 100 nm (Figure 12). The most pronounced change occurs at 70% relative humidity of the environment. This trend persists throughout the measurements, i.e., in the period of 14 min after atomization, and applies to the values of the mean diameter (CMD) as well as to the mode or median. A similar correlation occurs when the other characteristic grain size distributions are analyzed. For the medial value of grain size d50 (CMD), the increase in grain size as a function of time is smaller at about 30 nm. For the average value of nano grain size (CAD), the changes are similar, although additionally, a smaller dependence of grain size on moisture content is noted. In general, the lower the humidity of the atomization environment, the larger the grain sizes as a function of time. For median d_50_ values starting from humidity >70%, a stabilization at 150 nm is observed. For the mode size analysis, an increase in grain size was observed at 80% humidity.

The analysis of the characteristic sizes of aerosol grains as a function of humidity indicates a decrease in their size, with increasing ambient humidity values. This is especially clear when analyzing the mean value of grains according to the RRSB (Rosin-Rammler-Sperling-Bennet) model.

Due to the asymmetry of the grain distribution, it can be assumed that the RRSB function can be useful for the analysis of such an aerosol nanoparticle mixture, especially with distributions characterized by a high slant.

The cumulative curve of the RRSB function (after integration) is described by the following equation:(3)$$A_{f}=1-\exp{\left(\frac{-d}{d_{0.632}}\right)}^{a}$$
where:*A_f_*—cumulative grain size less than d*d*—grain diameter of the i-th fraction [m]*d_0.632_*—average population diameter by RRSB in which 63.2% of the grains are smaller (1 − 1/e = 0.632)*a*—constant of homogeneity

The uniformity constant a is the slope of the straight line ln[1/1 − *A_f_*)] with respect to d in logarithmic coordinates, since the linear form can be derived in Equation (3), while the ratio *d/d_0.632_* is the variance form of the nanoparticle size distribution. The convolution of the grain-frequency distribution function of the RRSB in the investigated size range, i.e., from 15–685 nm, can also represent the atomization activity of *A_f_* aerosol (neglecting the aspects of mass and shape of aerosol nanoparticles) in a large simplification (Figure 13).

The study showed that the value of the directional coefficient of the linear function changes with time towards higher values, so the granulometric distribution of the nanoparticles expands, especially at 90% humidity (from 0.2 to 0.5) (Figure 14). However, at a relative humidity of the atomization environment of 70%, the scattering of nanoparticles is the smallest and remains at a value of about 0.3. This means that two mechanisms can occur during atomization in a humid environment—aggregation of nanoparticles in a dry environment and clumping of aerosol nanoparticles in a very humid environment. It is worth noting that the confidence coefficient of changes in the directional coefficient of this function is at the level of 0.9 (90%) with the exception of the atomization environment with a relative humidity of 90% in which clumping of nanoparticles occurs (Figure 15).

## 4. Discussion

Fixed aerosol extinguishing systems are a generation of very effective systems used to extinguish all groups of fires, except for fire group D (combustible metals), although even here there is no conclusive evidence. Although the fire suppression mechanisms are not well understood, many successful implementations have been made using them.

Superfine powder particles act as fire suppressors through heterogeneous free radical inhibition. The heterogeneous (i.e., in different states of matter) quenching mechanism results in a ‘‘thermal death’’ effect of free radical flame burning (FFR) on the powder surface. The powder remains in the solid phase, does not evaporate and acts as a “cold medium” in which recombination reactions take place to break the chain combustion process. Each single particle of extinguishing powder, in the process of adsorption, contributes to the destruction of radicals, so it is important to have the maximum possible fineness, which can be obtained at the right humidity (70%).

Therefore, intensive research is still being conducted. First of all, it has been shown that firefighting aerosols are highly dispersed when atomized in an environment of about 70% humidity. This fact opens new cognitive possibilities towards the elucidation of their mechanism of action. Probably, water molecules and their radicals immediately adhere to the surface of atomized nanosized grains forming an envelope. Such a mechanism, on the one hand, hinders the aggregation of aerosol superfine particles caused by a large surface development, and on the other hand, effectively traps radicals. However, the aqueous envelope can evaporate at high temperatures and the aerosol becomes invisible. This probably occurred in the 70% humidity environment (Figure 3) where the high temperature in the test chamber caused the water envelope to evaporate. At lower temperatures, the aerosol can persist in the environment for a longer period of time.

Figure 16 shows a microphotograph of small grain agglomerates produced under wet conditions (70%). Single nano-grains with a diameter of 196.68 nm, for example, are also visible. In such an environment, the number of nano-grains, together with the captured radicals, is the highest. In other words, they can effectively hinder the necessary chain reactions that are the suggested basis for flame combustion.

A lower ambient humidity immediately changes the dispersion of aerosols toward larger sizes. This is caused by their rapid aggregation as in Figure 17. The size of such aggregates can be as small as tens of micrometers. Large aggregates are less effective in the radical mechanism of fire suppression.

## 5. Conclusions

In dry (30% humidity) and very humid (90% humidity) environments, the number of aerosol particles decreases over time due to both aggregation and clumping.The size of the mode fraction, median and mean value of the aerosol increases over time from 20 to 50 nm depending on the environmental humidity.The most preferable conditions for extinguishing aerosol atomization occur in an environment with a humidity of about 70%, which may be connected with more effective capture of water radicals of chain reactions occurring in flame fires.

## Figures and Tables

**Figure 1 materials-14-03329-f001:**
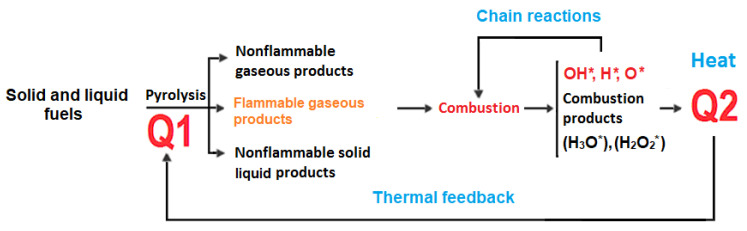
Scheme of combustion of solid and liquid fuels. Source: Own elaboration based on [1].

**Figure 2 materials-14-03329-f002:**
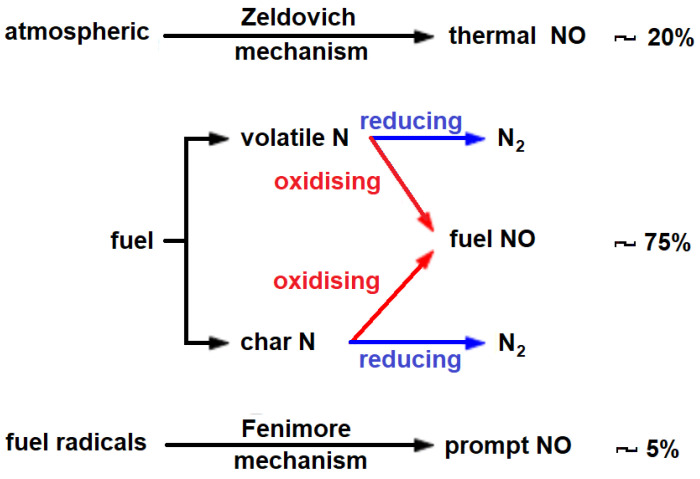
Scheme of nitrogen interaction in the generation of flame burning radicals.

**Figure 3 materials-14-03329-f003:**
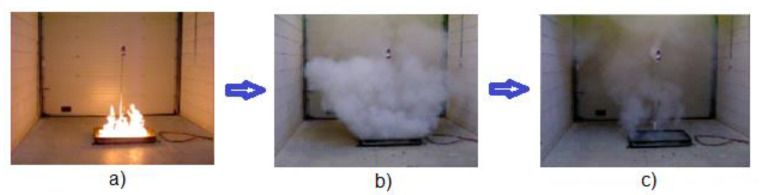
Test fire extinguishing (**a**) and aerosol visibility at 5 (**b**) and 15 s (**c**) after fire extinguishment in the test chamber.

**Figure 4 materials-14-03329-f004:**
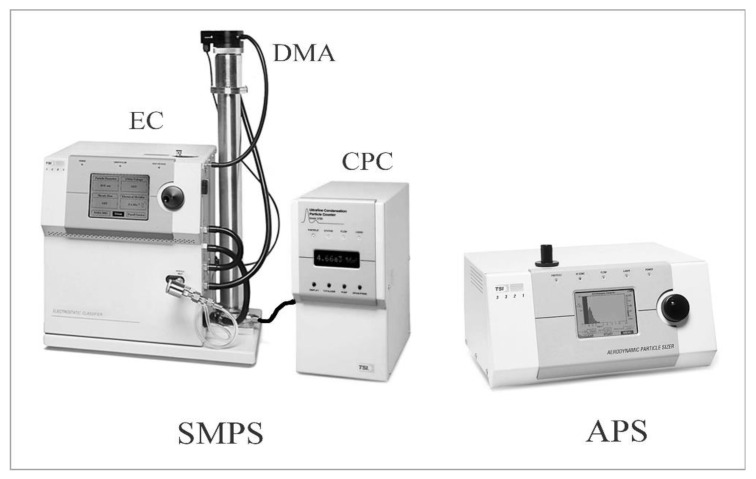
SMPS and APS particle spectrometers used in measurements.

**Figure 5 materials-14-03329-f005:**
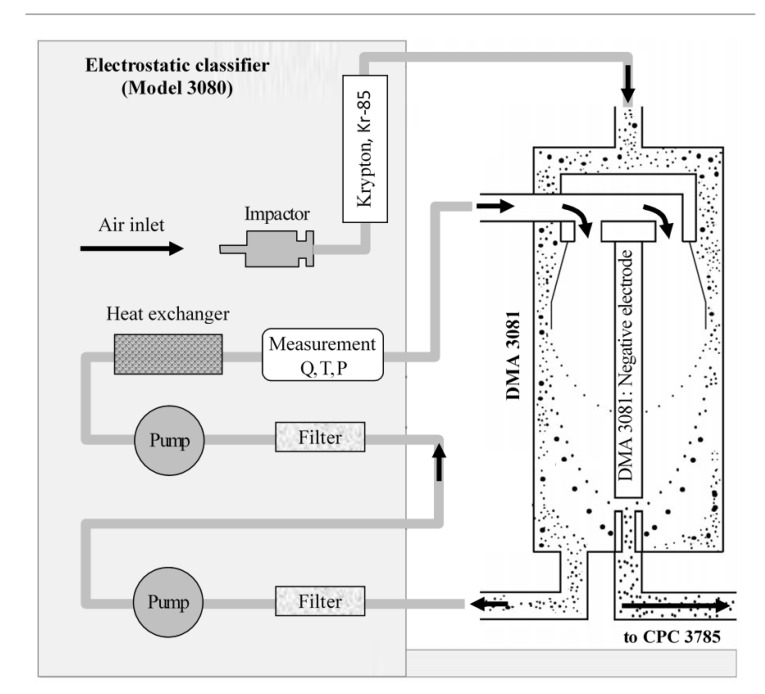
Classification of particles. The Model 3080 Electrostatic classifier (EC) scheme. (Q—flow rate, T—temperature, P—pressure).

**Figure 6 materials-14-03329-f006:**
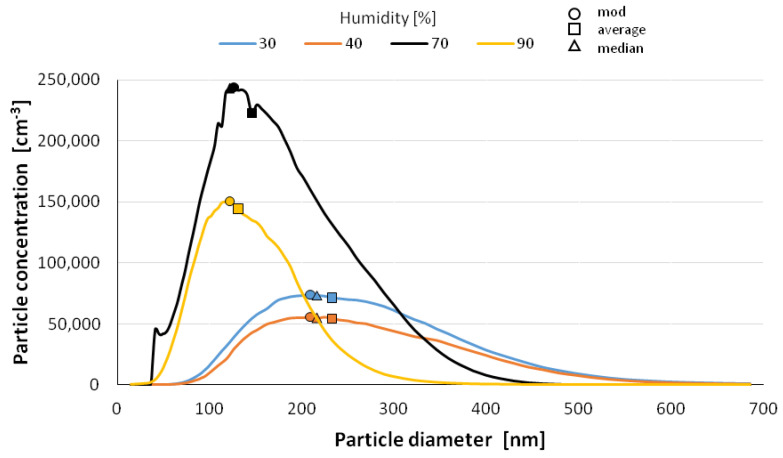
Size distribution of superfine particles 120 sec after atomization.

**Figure 7 materials-14-03329-f007:**
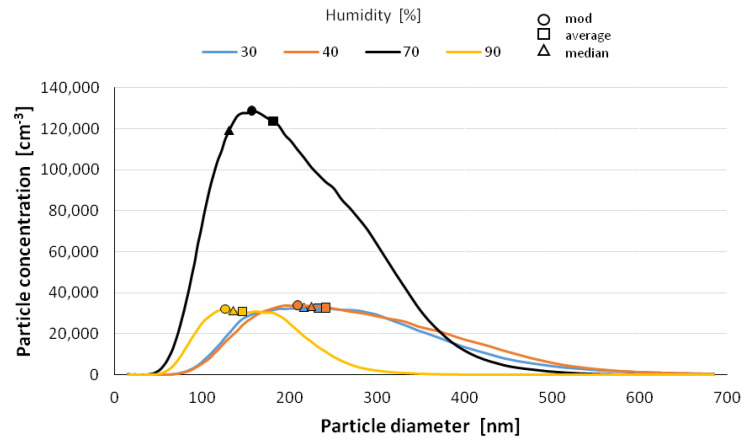
Size distribution of superfine particles 4.5 min after atomization.

**Figure 8 materials-14-03329-f008:**
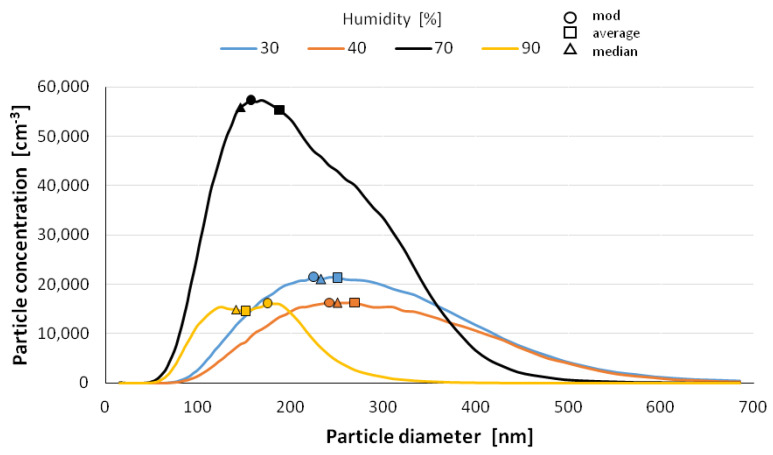
Size distribution of superfine particles 7.5 min after atomization.

**Figure 9 materials-14-03329-f009:**
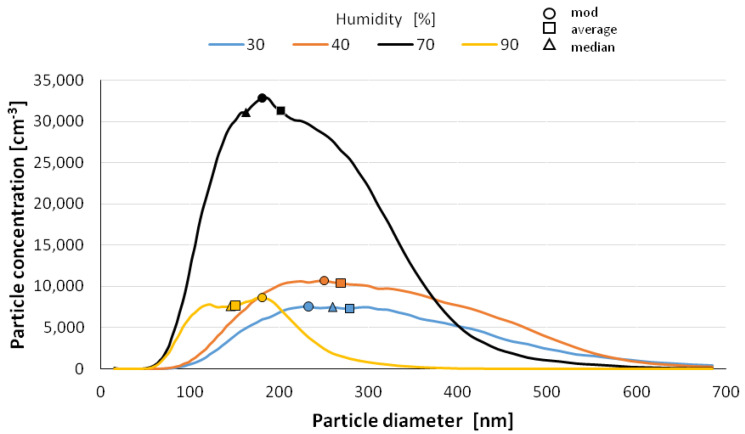
Size distribution of superfine particles 10.5 min after atomization.

**Figure 10 materials-14-03329-f010:**
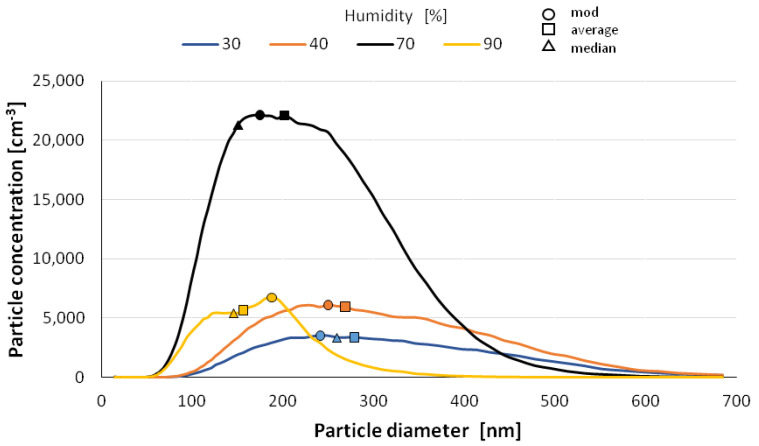
Size distribution of superfine particles 13.5 min after atomization.

**Figure 11 materials-14-03329-f011:**
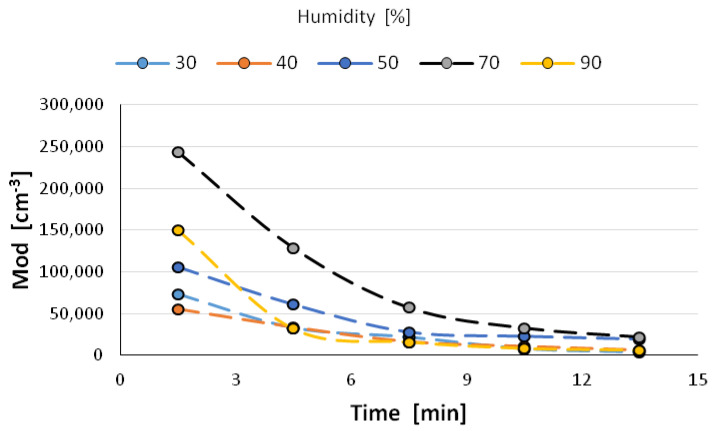
Change in nanoparticle abundance of the most common fraction.

**Figure 12 materials-14-03329-f012:**
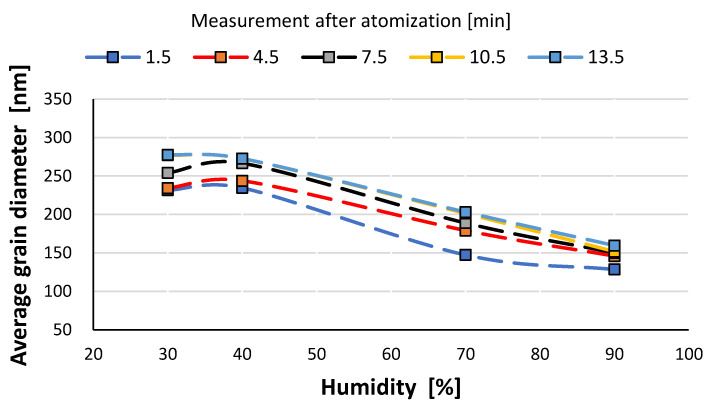
Change in the average size of nano-grain population as a function of humidity environment.

**Figure 13 materials-14-03329-f013:**
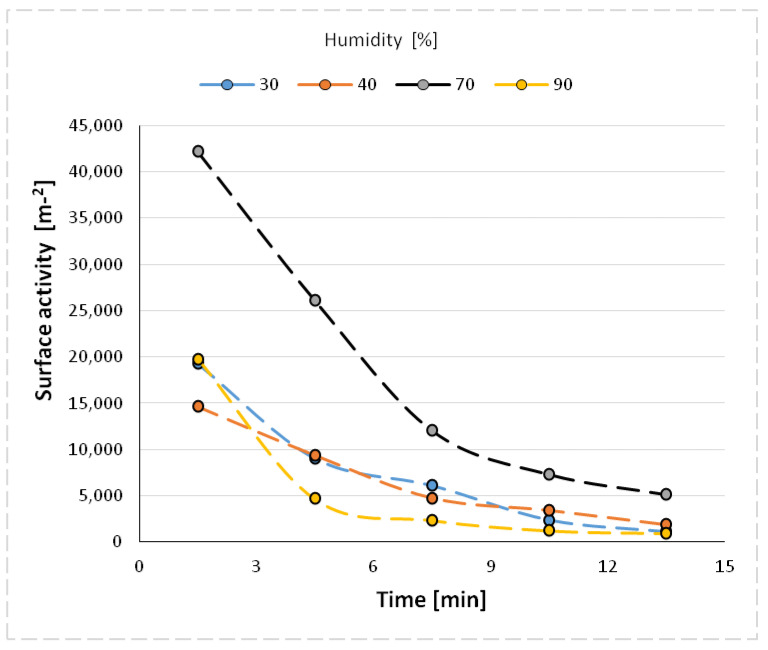
Aerosol atomization activity as a function of humidity and time.

**Figure 14 materials-14-03329-f014:**
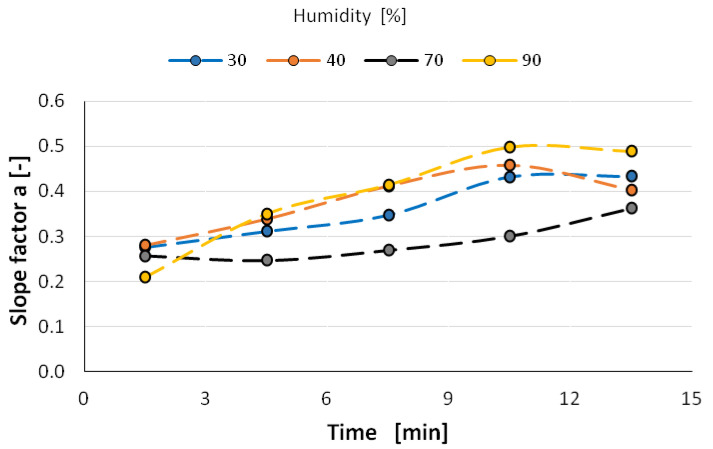
Directional coefficients of straights as a result of double logarithmic frequency functions RRSB.

**Figure 15 materials-14-03329-f015:**
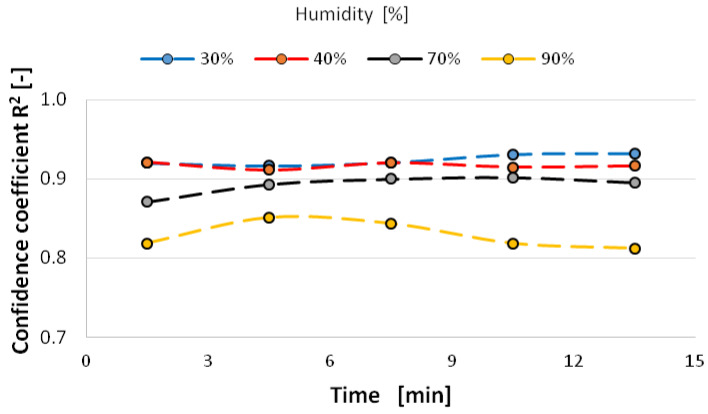
Confidence factor of the directional coefficient value as a function of time and ambient humidity during atomization.

**Figure 16 materials-14-03329-f016:**
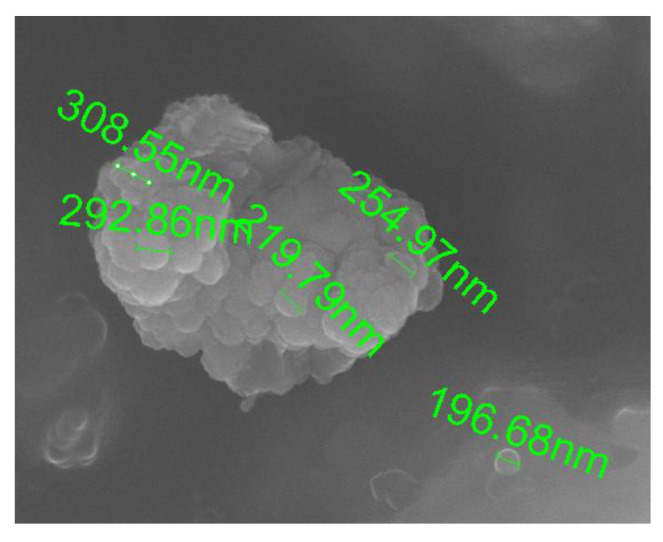
Smaller grain agglomerates and individual nanoparticles in a moist environment (humidity 70%). Samples were collected from the bottom of the test chamber.

**Figure 17 materials-14-03329-f017:**
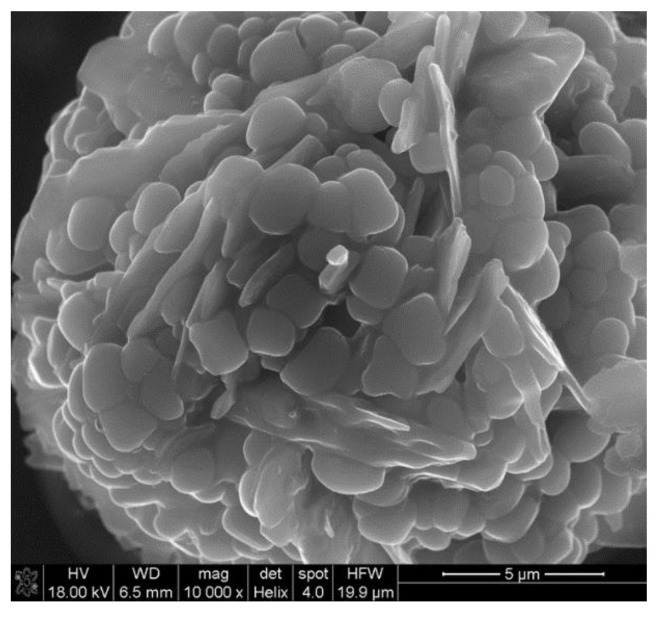
Aerosol grain agglomerates (approximately 30 m) obtained in a dry environment (humidity 30%). Samples were taken from the bottom of the test chamber.

## Data Availability

The data presented in this study are available on request from the corresponding author.

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
