# Peer review of "The Effect of Humidity on the Atomization Process and Structure of Nanopowder Designed for Extinguishment"

_materials, 2021, doi:10.3390/ma14123329_

Round 1

Reviewer 1 Report

The authors present a study on microparticles and their behavior under different relative humidity conditions. The manuscript is, for the most part, well-written, with a few grammatical/syntactical errors. However, the presentation has several areas that need improvement, particularly the figures. Specfic comments below:

Line 33: How are only coal and hydrogen the only fuels in the environment?

Figure 1 caption has two periods at the end of the first sentence.

Line 44: Sentence has two periods at the end.

In Section 2 (Research Significance) the authors should discuss the substantial growth of wildfires - both in number and intensity and explain how this research's relevance is augmented by this.

Figure 3: The "a)" and "b)" are grainy and the arrow is also low resolution.

Line 164: Superscript "3".

Line 174: The symbol inside the parentheses does not look formatted correctly.

Line 203: What is "fashion"?

Figures 5-9 do not seem to be made at the same aspect ratio and/or have the same font size. For example the title for Figures 5 and 6 ("Humidity") looks very different and Figure 6 looks shorter. Make sure all figures look similar.

Line 231: "starts" should be "started".

Line 251: There is a typo - it should read d<sub>0.632 (or equation 3 should be modified).

Line 254: Why is there a question mark following "w"?

Figure 12 does not need a dashed grey outline.

It is distracting to change the symbols (filled and unfilled circles) across figures - i.e. Figures 12, 13, 14.

Line 279: There are two periods at the end of the line.

There is a repeated Figure 14.

Figure 15 needs to be renumbered.

Conclusions are numbered as section 5 which is the same as Discussion.

Author Response

Response to Reviewer 1

The authors present a study on microparticles and their behavior under different relative humidity conditions. The manuscript is, for the most part, well-written, with a few grammatical/syntactical errors. However, the presentation has several areas that need improvement, particularly the figures. Specfic comments below:

 Thank you very much for the time and effort which let us to improve the quality of our article. The revision in the manuscript has been highlighted using yellow text.

Line 33: How are only coal and hydrogen the only fuels in the environment?

This is a simplification, of course there are many other fuels present in nature, coal, biomass, oil. In the text of the article we indicated that a simple assumption was made.

Figure 1 caption has two periods at the end of the first sentencje, and Line 44: Sentence has two periods at the end.

Thank you for reading the article very carefully. The sentences have been corrected.

In Section 2 (Research Significance) the authors should discuss the substantial growth of wildfires - both in number and intensity and explain how this research's relevance is augmented by this.

Thank you for your useful comment, the following text has been added to the article:

 “In recent years, there has been a clear increase in fires (13%). Among other things, climate change makes fires bigger, more intense and last longer than before. On the other hand, it is estimated (FAO) that humans are responsible for around 75% of all wildfires, and much of the increase in fire incidents during 2020 can be directly linked to human actions, both in terms of new technologies, materials and interventions. Science-based approach is needed to forecast risk and prioritize interventions, which are both critical elements in preventing fires before they need to be suppressed”

In addition, section "2" at the suggestion of the second reviewer was included in section 1 "introduction" and the entire article was renumbered

Figure 3: The "a)" and "b)" are grainy and the arrow is also low resolution.

Figure has been corrected, but it  has a low resolution because the images after 5 and 15 seconds were cut from the film, which was taken during the extinguishing of the model fire. Of course, we can share this movie available in full.

Line 164: Superscript "3".

Thank you for reading the article very carefully, superscript has been corrected

Line 174: The symbol inside the parentheses does not look formatted correctly.

Thank you very much for noticing, of course it was about beta β radiation, the fact that the co-authors are working on different versions of Word hence the formatting errors

Line 203: What is "fashion"?

Of course I meant "mode", the most frequently repeated value. Text corrected.

Figures 5-9 do not seem to be made at the same aspect ratio and/or have the same font size. For example the title for Figures 5 and 6 ("Humidity") looks very different and Figure 6 looks shorter. Make sure all figures look similar.

Thank you very much for this comment, the figures have been unified

Line 231: "starts" should be "started".

Thank you for notice.

Line 251: There is a typo - it should read d<sub>0.632 (or equation 3 should be modified).

Of course there was a mistake, thank you, text corrected

Line 254: Why is there a question mark following "w"?

Error in the text, it has been deleted

Figure 12 does not need a dashed grey outline.

As suggested by the reviewer, the outline  has been erased

It is distracting to change the symbols (filled and unfilled circles) across figures - i.e. Figures 12, 13, 14.

Thank you very much for your attention, all figures have been corrected

Line 279: There are two periods at the end of the line.

Thank you for reading the article very carefully. The sentence has been corrected

There is a repeated Figure 14. and Figure 15 needs to be renumbered.

Figures have been renumbered

Conclusions are numbered as section 5 which is the same as Discussion.

As recommended by the reviewer, the sections have been renumbered

Reviewer 2 Report

First I kindly appreciate your time and effort for this study.

Let me ask you several questions and recommend a few things regarding this paper as belows.

1) It seems better that ch. 2 should be included into ch .1.

2) In ch. 3.2, a specifications table and a schematic diagram of measuring equipments  should be presented.

3) In ch. 4 Results, there is not enough explanation about the mechanism of effect of humidity on the particle size and concentration.

    - Why does the particle size with the max. concentration decrease at high humidity?

    - Why is the particle size with the max. concentration increasing in accordance with elapsed time.

    - What is the reason that particle concentration at a humidity of 30% is higher than 40% until 7.5min but it reverses after that.

4) In the Fig. 7~9, the line color of humidity 40% is changed to Red from Orange. Please correct them. And also, change the line color of 13.5min in the Fig. 11 into other one as it is confused with the line of 1.5min.

5) Since the particle concentraion drops down at a relative humidity of 90%, we cannot know the exact critical humidity value where the concentration changes. So I strongly recommend to obtain the results at a relative humidity of 80% additionally.

6) In the fig. 14 and 15, you need to insert microphotos of another humidity for easy comparison.

7) The chapter number of 'Conclusions' is 6 not 5.

8) The 3rd conclusion doesn't seem appropriate because no extinction test is conducted in your study and it a kind of guess.

Author Response

Thank you very much for the time and effort which let us to improve the quality of our article. The revision in the manuscript has been highlighted using yellow text .

All responses and recommendations are included in the attachment

Reviewer 3 Report

The comments are attached herewith. 

Round 2

Reviewer 2 Report

I think that enough revision has been done.

I appreciate your time and effort.

Author Response

Thank you.

Reviewer 3 Report

Accept can be suggested.